# Prediction of Chemoresistance—How Preclinical Data Could Help to Modify Therapeutic Strategy in High-Grade Serous Ovarian Cancer

Jacek Wilczyński [1,*], Edyta Paradowska [2], Justyna Wilczyńska [3] and Miłosz Wilczyński [4,5]

1   Department of Gynecological Surgery and Gynecological Oncology, Medical University of Lodz,
    4 Kosciuszki Str., 90-419 Lodz, Poland
2   Laboratory of Virology, Institute of Medical Biology of the Polish Academy of Sciences, 106 Lodowa Str.,
    93-232 Lodz, Poland; eparadow@cbm.pan.pl
3   Department of Tele-Radiotherapy, Mikolaj Kopernik Provincial Multi-Specialized Oncology and
    Traumatology Center, 62 Pabianicka Str., 93-513 Lodz, Poland; justinawilczynska@gmail.com
4   Department of Gynecological, Endoscopic and Oncological Surgery, Polish Mother's Health
    Center—Research Institute, 281/289 Rzgowska Str., 93-338 Lodz, Poland; milosz.wilczynski@iczmp.edu.pl
5   Department of Surgical and Endoscopic Gynecology, Medical University of Lodz, 4 Kosciuszki Str.,
    90-419 Lodz, Poland
*   Correspondence: jrwil@post.pl

**Abstract:** High-grade serous ovarian cancer (HGSOC) is one of the most lethal tumors generally and the most fatal cancer of the female genital tract. The approved standard therapy consists of surgical cytoreduction and platinum/taxane-based chemotherapy, and of targeted therapy in selected patients. The main therapeutic problem is chemoresistance of recurrent and metastatic HGSOC tumors which results in low survival in the group of FIGO III/IV. Therefore, the prediction and monitoring of chemoresistance seems to be of utmost importance for the improvement of HGSOC management. This type of cancer has genetic heterogeneity with several subtypes being characterized by diverse gene signatures and disturbed peculiar epigenetic regulation. HGSOC develops and metastasizes preferentially in the specific intraperitoneal environment composed mainly of fibroblasts, adipocytes, and immune cells. Different HGSOC subtypes could be sensitive to distinct sets of drugs. Moreover, primary, metastatic, and recurrent tumors are characterized by an individual biology, and thus diverse drug responsibility. Without a precise identification of the tumor and its microenvironment, effective treatment seems to be elusive. This paper reviews tumor-derived genomic, mutational, cellular, and epigenetic biomarkers of HGSOC drug resistance, as well as tumor microenvironment-derived biomarkers of chemoresistance, and discusses their possible use in the novel complex approach to ovarian cancer therapy and monitoring.

**Keywords:** ovarian cancer; prediction; biomarkers; chemoresistance



## 1. Introduction

High-grade serous ovarian cancer (HGSOC) is the most lethal tumor of the female genital tract due to lack of screening programs for the average-risk population, followed by the delayed diagnosis. It has also a high proliferative potential and recurrence rate. Therefore, the 5-year survival in the advanced patient population (clinical stages III–IV) is unsatisfactory, although recently this has been improved by the introduction of poly-ADP-ribose polymerase (PARP) inhibitors in the group of homologous recombination deficiency (HRD)-positive tumors (data of American Cancer Society 2020. https://www.cancer.org/cancer/ovarian-cancer/detection-diagnosis-staging/survival-rates.html, accessed on 10 August 2023) [1,2]. The approved standard therapy consists of primary cytoreductive surgery followed by standard platinum/taxane-based chemotherapy or,

alternatively, in inoperable tumors, of neo-adjuvant platinum/taxane chemotherapy followed by interval cytoreduction. In the case of advanced and sub-optimally operated tumors, anti-VEGF (vascular–endothelial growth factor) humanized monoclonal antibody bevacizumab has been approved, while, in the group of HRD-deficient patients, the poly-ADP ribose polymerase (PARP) inhibitors are used, with PARP-based therapy showing satisfactory efficacy [2–8]. In phase III, PAOLA-1/ENGOT-ov25 trial patients with the *BRCA*-mutated tumors benefitted from maintenance therapy using the PARP inhibitor olaparib + bevacizumab versus placebo + bevacizumab (24-month progression-free survival PFS 89% vs. 15%) [9]. In the VELIA study, complete response rates among patients' residual disease after cytoreduction were 24% in the PARP inhibitor veliparib group compared with 18% in the controls [10]. The PRIMA/ENGOT-ov26/GOG-3012 trial showed that, in newly diagnosed advanced ovarian cancer, therapy with the PARP inhibitor niraparib was effective compared with placebo irrespectively of HRD status (HRD-positive 38% vs. 17%, overall 24% vs. 14%) [8]. However, resistance to PARP inhibitors has unfortunately been observed [11]. High-grade serous ovarian cancer is chemo-responsive to the first line of standard platinum and taxane-based chemotherapy, particularly in the group of *BRCA*-mutated patients. However, the main therapeutic problem appears when chemotherapy is used to treat the patent or recurrent disease, as well as in the cases of primary chemo-refractoriness. In these cases, treatment is eventually ineffective and the disease is usually lethal. Therefore, identification of the biomarkers of drug resistance is one of the most desirable activities in ovarian cancer surveillance and therapy. Chemoresistance is regulated by many different mechanisms originating both from the cancer cells themselves and from the tumor microenvironment (TME). Recent studies have brought much information about genetic HGSOC heterogeneity, disturbed epigenetic regulation, and the modulating role of TME. Tumors of different genetic or epigenetic signature and TME composition are most probably characterized by diverse drug responses [12–15]. The prediction concerning the drug resistance of the primary, metastatic, and recurrent tumors should be an indispensable element of the personalized anti-cancer therapy. As chemosensitivity could change depending on both localization (different TME) and time in the course of the disease, repetitive monitoring of the chemosensitivity during therapy should be another step in the management of HGSOC patients. The review presents the spectrum of biomarkers from the tumor itself and its TME, for prediction of chemoresistance in HGSOC, and discusses the new possibilities of planning and monitoring therapy.

## 2. Analysis of Cancer Tissue-Derived and Peripheral Blood Biomarkers

### 2.1. Genetic and Proteomic Biomarkers

The problem of chemoresistance concerns not only the chemotherapeutics but also drugs used in the targeted therapy. The classic biomarkers of platinum and PARP sensitivity are the germinal and somatic mutations of *BRCA1/2* genes and the HRD status of the tumor, respectively [16]. The reverse mutations in *BRCA* genes are able to switch chemosensitivity into chemoresistance. *BRCA* reverse mutations were identified in cfDNA of 18% of pre-treatment patients with platinum-refractory HGSOC cancer and in cfDNA of 13% of platinum-resistant tumors. Patients without reverse mutations had better outcomes during rucaparib therapy [17]. Reverse mutations in other homologous recombination repair (HR) genes like *RAD51C, RAD51D,* and *PALB2* are also responsible for secondary resistance to platinum- and PARP inhibitor-based therapy [17,18]. The highly expressed PARP, Fanconi Anemia complementation group D2 (*FANCD2*) and p53 proteins are the feature of BRCAness phenotype and are positively correlated with platinum resistance [19]. The drug-resistant and drug-sensitive recurrent ovarian cancers were shown to possess unique genetic alterations when studied by ctDNA liquid biopsy. *TP53* was the most frequently mutated gene in both groups. Copy number variations of *MYC*, *RB1*, and phosphatidylinositol-4,5-bisphosphate 3-kinase catalytic subunit alpha (*PIK3CA*) were noticed in recurrent cancers, while *BRCA2* N372H polymorphism was noticed in recurrent drug-resistant tumors [20]. Whole-genome sequencing (WGS) analysis of ctDNA in

heavily pre-treated HGSOC patients confirmed that the most mutated gene was *TP53* (94% of HGSOC cases). The WGS detected mutated *TP53* ctDNA in 88% of cases, but there was a low correlation between plasma and tumor copy number alteration profiles; however, it was high in the subgroup with the highest ctDNA tumor fraction. Both the genome-altered fraction and plasma mutation burden had prognostic value according to chemoresistance and survival [21]. More than half of HGSOC have a defective HR pathway, but in tumors with intact HR, an amplification or gain of cyclin E1 (*CCNE1*) was observed and was related to chemoresistance and decreased OS [22]. Based on the HRD score, gene insertions and deletions, copy number changes load, duplication load, single nucleotide variants, and mutational signatures, a predictor of platinum-resistance, named DRD score, were established and validated in a cohort of HGSOC patients reaching the sensitivity of 91% [23]. Cytogenetic analysis indicated that several losses (13q32.1 and 8p21.1, 8p and 9q) or gains (1q, 5q14~q23, and 13q21~q32, 9p13.2–13.1, 9q21.2–21.32, 9q22.2–22.31, 9q22.32–22.33, and 9q33.1–34.11) in chromosomal regions were connected to chemoresistance [24–27]. The presence of several SNPs, between others, rs4910232 (11p15.3), rs2549714 (16q23), and rs6674079 (1q22), was also connected to poor response for first-line platinum-based chemotherapy and unfavorable outcome in ovarian cancer patients [28]. Mutations in the members of the A disintegrin and metalloproteinase with thrombospondin motifs (ADAMTS) family were associated with a significantly higher chemosensitivity (100% for ADAMTS-mutated vs. 64% for ADAMTS wild-type cases), and with significantly better OS and PFS [29]. In the patient-derived xenograft mouse model of ovarian cancer, the differential expression of Sin3A-associated protein 25 (*SAP25*), major histocompatibility complex, class II, DP alpha 1 (*HLA-DPA1*), AKT serine/threonine kinase 3 (*AKT3*), and phosphoinositide-3-kinase regulatory subunit 5 (*PIK3R5*) genes and mutation of transmembrane protein 205 (*TMEM205*) and DNA-directed RNA polymerase II subunit RPB1 (*POLR2A*) coding genes were found to have unfavorable function in the induction of chemoresistance [30]. Conversely, transcriptome analysis identified interferon regulatory and tumor suppression factor 1 (IRF1) transcription factor as a supporter of platinum sensitivity in HGSOC [31]. Study of gene networks and expression of quantitative trait loci (eQTLs) indicated that many of the mapped genes associated to chemoresistance in ovarian cancer were located in chromosome 9, which supported the previous observations of the connection between gene alterations in that region with progression and chemoresistance of ovarian tumors [32,33]. A total of 96% of genes were co-regulated by the same transcription factor organic cation transporter 1 (OCT1) which is engaged in disturbances of platinum-induced apoptosis. Another important protein was valosin-containing protein (VCP) which played a critical role in the disintegration of polypeptide cellular structures. Low expression of VCP is found in ovarian cancer cells, especially from platinum-resistant cell lines and ovarian cancer cohorts [32,34]. Other genes which correlated to chemoresistance were *BRCA2* and neighboring NEDD4 binding protein 2-like 1 (*N4BP2L1*), -like 2 (*N4BP2L2*), FRY microtubule binding protein (*FRY*), and StAR related lipid transfer domain containing 13 (*STARD13*) genes [26]. The *BRCA2* and *STARD13* are tumor-suppressors, while up-regulation of *N4BP2L1* and *N4BP2L2* was associated with positive prediction in ovarian cancer patients [35]. The up-regulation of *BRCA2* was observed in chemotherapy resistant patients, and the down-regulation of *BRCA2* reduced the DNA repair in ovarian cancer cells, sensitizing them to cisplatin [36]. The chemo-refractory HGSOC genotype is also characterized by other exclusive genomic and transcriptional modifications represented by changed expression of several genes, including: hypoxia-induced factors (*HIF*), tumor necrosis factor (*TNF*), JUN transcription factor (*JUN*), FOR proto-oncogene family (*FOS*), growth arrest and DNA damage inducible beta (*GADD45B*), induced myeloid leukemia cell differentiation anti-apoptotic protein Mcl-1 (*MCL1*), C-X-C motif chemokine receptor 4 (*CXCR4*), snail family zinc finger 1 (*SNA1*), vimentin (*VIM*), soluble N-ethylmaleimide-sensitive factor-attachment protein (SNAP) receptors *SNAREs*, member of RAS oncogene family (*Rab*), and NF-κB transcription factor [37–39]. The genomic and translational changes that drive to acquired chemoresistance in HGSOC

are represented by disturbed expression of multi drug resistance 1 (*MDR1*), glutathione S-transferase Pi 1 (*GSTpi*), B-cell lymphoma 2 (*BCL-2*), survivin, SMAD family member 4 (*SMAD4*) and *β*-tubulin III, and CpG methylation [16,40–44]. High expression of two genes angiogenic factor with G-Patch and FHA domains 1 (*AGGF1*), and microfibril associated protein 4 (*MFAP4*), was observed in HGSOC and correlated with platinum resistance [45]. In advanced stage HGSOC, stemness-associated genes were found to be connected to the resistance to platinum and to combined platinum-taxol therapy. The expression of Aurora A kinase-*AURKA*, cyclin A2-*CCNA2*, MYB proto-oncogene-like 2-*MYBL2*, and origin recognition complex subunit 1-*ORC1* affected the survival of platinum resistant patients, while the expression of *CCNA2*, cyclin-dependent kinase 1-*CDK1*, *ORC1*, DNA topoisomerase II alpha-*TOP2A*, and threonine tyrosine kinase-*TTK* affected the survival of platinum/taxol-resistant patients [46]. The problem of chemoresistance also concerns the group of patients subjected to neo-adjuvant chemotherapy. The highest incidence of copy number variations was found in three genes: maestro heat-like repeat-containing protein family member 1 (*MROH1*), transmembrane protein 249 (*TMEM249*), and heat shock transcription factor 1 (*HSF1*), and was associated with neo-adjuvant chemoresistance [47].

The chemoresistant group and the chemosensitive group showed differentially expressed plasma proteins. Among them, complement C4-A, IgJ chain, clusterin, *α*-1-antitrypsin, and carbonic anhydrase 1 were up-regulated, and transthyretin, haptoglobin, β-2-glycoprotein, Ig γ-2 chain C region, Ig γ-1 chain C region, complement factor I light chain, Igκ chain C region, complement C3, and apolipoprotein E were down-regulated in the chemoresistant group when compared with the chemosensitive group [48]. The shed form of cellular desmoglein-2 (DSG2), an epithelial junction protein, was significantly over-expressed in the serum of HGSOC patients with chemo-refractory cancer and worse survival. Shedding of DSG2 is mediated by EGFR followed by a MMP cleavage and accompanies many malignancies [49]. Soluble programmed death receptor ligands sPD-L1 and sPD-L2 were related to reduced OS and platinum resistance, respectively [50]. Secretome proteomics of chemo-resistant and chemo-sensitive ovarian cancer enabled the identification of the proteins that showed different levels between the studied groups. One of them, collagen type XI alpha 1 chain (COL11A1), was found to be a biomarker of chemoresistance and correlated with worse survival [51]. The flow cytometric analysis of the blood of ovarian cancer patients found significant relationship between chemoresistance and both increased the number of CD44+/CD24− stem cells and over-expression of ubiquitin-conjugating enzyme E2 coding gene RAD6 [52]. The population of CD44+/CD117+ stem cells were related to platinum and paclitaxel resistance [53].

Transcriptome-based stemness-related gene signature was used to create the prediction system for platinum sensitivity. The results revealed that four genes were associated with the OS of advanced-stage HGSOC patients, with *AURKA*, *MYBL2*, and *ORC1* predicting shorter survival, and Polo-like kinase 1-*PLK1* longer survival, respectively. The study performed to investigate the potential chemoresistance indicators among the stemness-associated key genes in stage III–IV HGSOC patients receiving platinum or the combination of platinum and taxol showed that the higher expression of *AURKA*, *MYBL2*, and *ORC1* genes prognosed shorter OS, while the higher expression of *CCNA2* gene prognosed longer OS. According to the PFS, the higher expression of *AURKA*, *BIRC5*, *CCNA2*, *CCNB2*, *CDC20*, *CDK1*, *PLK1*, *RRM2*, *TOP2A*, and *TTK* predicted longer PFS. In the group of advanced HGSOC patients treated with both platinum and taxol, shorter OS was correlated with the higher expression of *CCNA2*, *CDK1*, *ORC1*, *TOP2A*, and *TTK*, whereas higher expression of *BIRC5*, *CCNB2*, *CDC20*, *MYBL2*, *PLK1*, *TOP2A*, and *TTK* predicted longer PFS. The expression of *BIRC5*, *BUB1*, *CDC20*, *CDK1*, and *ORC1* was up-regulated in platinum-sensitive compared with platinum-resistant HGSOC samples. From all these genes, *CDC20* was found to be the most relevant gene for tumor progression and drug resistance [46].

## 2.2. Circulating Tumor Cells (CTCs)

The presence of particular populations of CTCs was also considered a prognostic factor for chemo-resistance. Analysis of EpCAM, mucin (MUC1 and MUC16), and excision repair cross-complementation group 1 (ERCC1) protein positive CTCs indicated that ERCC1+ CTCs were independent prognostic factors for platinum resistance, OS, and PFS in primary ovarian epithelial malignant tumors [54]. The presence of ERCC1+ CTCs correlated with sPD-L2 serum levels, and their persistence, indicated poor post-treatment outcome [50,55]. A stronger concordance of platinum sensitivity was noted for elevated iCTCs than for serum CA125 [56]. Upon platinum-based chemotherapy, CTCs acquired EMT-like phenotype, characterized by a shift towards PI3Kα and Twist-expressing CTCs, which could reflect a clonal tumor evolution towards therapy-resistant phenotype [57]. The analysis of CTCs for prognostic purposes is available even on a level of single CTCs. The CTCs positive for both stem cell (CD44, ALDH1A1, Nanog, Oct4) and EMT markers (N-cadherin, Vimentin, Snai2, CD117, CD146) could account for chemoresistance; however, the interpretation of the results is hampered by an inter-cellular and intra/inter-patient heterogeneity [58].

## 2.3. Epigenetic Biomarkers

### 2.3.1. Histone and Methylation Biomarkers

Several studies identified some mechanisms of epigenetic regulation of chemoresistance in HGSOC. Gene sets associated with H3K27me3/H3K4me3 histone marks at transcription start sites in a HGSOC tumor were investigated in one of the studies. The significantly lower expression for the H3K27me3 and bivalent gene sets in "stem-like cells" and in the chemoresistant cell lines made the point of the role of genetic silencing in HGSOC progression [59]. Another study revealed that by recruiting the DOT1-like histone lysine methyltransferase (DOT1L), CCAAT/enhancer-binding protein beta (C/EBPβ) can maintain an open chromatin state by H3K79 methylation of multi-drug resistance genes, thereby augmenting the chemoresistance of tumor cells [60]. 5-Hydroxymethylcytosine (5hmC) may regulate gene expression or prompt DNA methylation. Loss of 5hmC levels have been associated with resistance to platinum-based therapy and worse patient survival [61]. The aberrant miR-7 methylation followed by changed regulation of MAF BZIP transcription factor G (*MAFG*) target gene has been involved in the development of platinum resistance and was associated with poor prognosis in ovarian cancer patients [62]. DNA/RNA helicase Schlafen-11 (SLFN11) is one of the strongest predictors of sensitivity to platinum. In tumor-infiltrating immune cells, SLFN11 expression was associated with immune activation in HGSOC by platinum treatment. However, CpG island hypermethylation of SLFN11 promoter was associated with platinum resistance in HGSOC patients [63,64]. Hypomethylating agents are able to resensitize tumor cells to cisplatin [65]. However, hypomethylation could also enhance the expression of plasminogen activator inhibitor type 1 (SERPINE1) and EMT in ovarian cancer, thus supporting chemoresistance [66].

### 2.3.2. MicroRNA Biomarkers

MiRNAs are known regulators of cancer gene's expression and function, and several miRNAs were found to play a prognostic and predictive role in ovarian cancer therapy. MiR-335 expression level was observed to be reduced in malignant tissue samples, especially omental implants, and was associated with shorter OS and tumor recurrence [67]. Up-regulation of miR-335-5p expression restored the cisplatin sensitivity of ovarian cancer cells through suppressing the *BCL2L2* anti-apoptotic gene, suggesting the potential of miR-335-5p/BCL2L2 signaling as a therapeutic target to overcome the cisplatin resistance [68]. High collagen type XI alpha 1 (COL11A1) levels were found to be associated with tumor progression, chemoresistance, and poor patient survival. MiR-509 and miR-335 were identified as the candidate miRNAs regulating COL11A1 expression. Treatment of ovarian cancer cells with miR-335 mimics decreased COL11A1 expression and suppressed cell proliferation and invasion, simultaneously increasing the cisplatin sensitivity [69]. Moreover, COL11A1 regulates twist family basic helix–loop–helix transcription factor 1-

related protein 1 (TWIST1), resulting in the induction of chemoresistance and inhibition of apoptosis in ovarian cancer cells [70].

The set of four miRNA biomarkers (miR-454-3p, miR-98-5p, miR-183-5p, and miR-22-3p) identified in the ovarian tumor were able to discriminate between platinum-sensitive and platinum-resistant HGSOC patients, thus being an indicator of chemoresistance [71]. The set of ten miRNAs (miR-151, miR-301b, miR-505, miR-324, miR-502, miR-421, let-7a, miR-320, miR-146a, and miR-193a) was used to create a 10-miRNA score. The tumor samples were classified into four subtypes: mesenchymal (high expression of stromal components), proliferative (high expression of proliferation markers), immunoreactive (high expression of T-cell chemokine ligands and their receptors), and differentiated (high expression of ovarian tumor markers) [72]. Results indicated that the high 10-miRNA-score group contained more tumors of the proliferative subtype, while the low 10-miRNA-score group contained more tumors of the mesenchymal subtype, respectively. A high 10-miRNA score was associated with extreme genome instability that explained the chemosensitivity, followed by favorable survival [61]. Low tumor suppressor miR-let7g tissue levels in ovarian cancer were associated with chemoresistance, and up-regulation of this miRNA in ovarian cancer cell lines promoted cell cycle arrest, inhibited epithelial-to-mesenchymal transition (EMT), and restored the chemosensitivity [73]. The PCR analysis confirmed the up-regulation of another miRNA, miR-23a-3p, in chemo-resistant tumors from HG-SOC patients. MiR-23a-3p suppresses apoptosis of tumor cells and supports platinum chemoresistance by regulating the expression of the apoptotic protease-activating factor 1 *APAF1*. The miR23a-3p/*APAF1* signaling could be a possible target to reverse platinum resistance [74]. Restoration of miR-139-5p in chemo-resistant ovarian cancer cell lines increased the sensitivity to cisplatin treatment and promoted cisplatin-induced mitochondrial apoptosis [75]. Lipid metabolism is important for sustainment of proliferation and stemness of ovarian cancer cells. Solute carrier family 27 member 2 (SLC27A2) is responsible for transporting long-chain and very long-chain fatty acids into the cells and activating intracellular signaling pathways. SLC27A2 can bind to the miR-411 promoter region and change its effects on the drug-transporter *ABCG2* target gene. By this mechanism, down-regulated miR-411 expression contributes to ovarian cancer chemoresistance [76]. The epigenetic regulation of insulin growth factor-1 receptor (IGF1R) by another miR-1294 has also been connected to cisplatin resistance [77]. Study of both primary and recurrent BRCA1/2-mutated ovarian cancers, and patient-derived cell lines used in the in vivo BRCA2-mutated mouse model, allowed for identification of miR-493-5p that induced platinum/PARPi resistance exclusively in BRCA2-mutated tumors [78]. Disturbed function of ion channels is one of the features of cancer cells. Expression of potassium channel calcium activated large conductance subfamily M alpha member 1 (KCNMA1) is reduced in chemoresistant ovarian cancer cells and correlates with the over-expression of miR-31 regulatory miRNA [79]. MiRNA-21 is an oncogenic miRNA found to be up-regulated in almost all human cancers. The miR-21 expression was up-regulated in cisplatin-resistant compared with cisplatin-sensitive ovarian cancer cells, and its expression was regulated by c-Jun N-terminal protein kinase 1 (JNK-1)/c-Jun/miR-21 pathway [80]. In vitro experiments indicated that miR-551b enhanced the proliferation and chemoresistance of ovarian cancer stem cells through the suppression of forkhead box O3 (*Foxo3*) and tripartite motif containing 31 (*TRIM31*) tumor suppressor genes [81]. However, there is also a group of miRNAs in which over-expression was correlated with increased chemosensitivity, like miR-9, miR-30a, and miR-211 [82–84]. The increase in the cancer stem cells population results from the activation of the Hippo/YAP pathway target genes upon myosin phosphatase target subunit 1 (*MYPT1*) down-regulation, mediated by the over-expression of miR-30b. Combination therapy with cisplatin and YAP inhibitors could potentially suppress *MYPT1*-induced resistance [85]. Defective function of the DNA repair and genome integrity checkpoints is responsible for the genetic instability of cancer cells. The Chk1 is a serine/threonine kinase which is activated in response to diverse genotoxic signals and retransmits signals from the proximal checkpoint kinases like ATM serine/threonine

kinase (ATM), ATR serine/threonine kinase (ATR), and serine/threonine kinase (ATX). The consequence of the signal transmission depends on the route of further signals and could be as follows: a switch to the stress-induced transcription program, initiation of DNA repair, delay or sustained block of cell cycle progression, apoptosis, and modulation of the chromatin remodeling pathway [86]. Dual oxidase maturation factor 1 (DUOXA1) over-expression stimulates reactive oxygen species (ROS) production and activates the ATX-Chk1 pathway [87]. The ROS production and DUOXA1 up-regulation cause the activation of c-Myc/miR-137/enhancer of zeste 2 polycomb repressive complex 2 subunit methylotransferase (EZH2) pathway and enhances platinum resistance [88]. Combination of the EZH2 inhibitor with a RAC1 GTPase inhibitor reduced the expression of stemness and induced inflammatory gene expression, thus promoting the differentiation of subpopulations of HGSOC cells and chemosensitivity [89].

### 2.3.3. Long Non-Coding RNA Biomarkers

The panel of seven HGSOC tumor-derived long non-coding RNAs (lncRNAs) including both up-regulated (RP11-126K1.6, ZBED3-AS1, RP11-439E19.10, and RP11-348N5.7) and down-regulated lncRNAs (RNF144A-AS1, GAS5, and F11-AS1) showed high accuracy in predicting chemosensitivity (AUC > 0.8) [90]. Another panel of seven lncRNAs was also shown to predict chemoresistance with LINC01363, AC114401.1, and AL360169.2 indicating chemo-resistance, and LINC01018, LINC02636, AC090625.2, and AC084781.1 indicating at least partial chemosensitivity [91]. Plasmacytoma variant translocation 1 (PVT1) is a lncRNA responsible for dysregulation and down-regulation of tumor suppressors. The PVT1 was significantly up-regulated in ovarian cancer tissues of cisplatin-resistant patients, acting through the regulation of TGF-β1, p-Smad4, and Caspase-3 expression in apoptotic pathways [92]. Metastasis-associated lung adenocarcinoma transcript 1 (MALAT1) regulates the expression of metastasis-associated genes in cancer. Its knockdown enhanced platinum-induced apoptosis in vivo and inhibited the Notch1 signaling pathway and ATP binding cassette subfamily C member 1 (ABCC1) drug transport system expression in platinum-resistant ovarian cancer cells [93]. HOX transcript antisense RNA (HOTAIR) lncRNA is a regulator of chromatin state and is over-expressed in several metastatic tumors. It was found in ovarian cancer that the NF-κB-HOTAIR axis contributes during platinum-triggered DNA damage response to cellular senescence and chemotherapy resistance [94].

Long integrated non-coding RNAs (lincRNAs) play an important role in platinum-induced DNA-damage response. LincRNA H19 knockdown in the HGSOC cell line resulted in the recovery of cisplatin sensitivity through the reduction of seven key proteins involved in the glutathione metabolism pathway [95].

### 2.3.4. Circular RNA Biomarkers

Circular RNAs (circ_RNAs) are responsible for the sponging of miRNAs, a process which could both promote or suppress the proliferation and invasiveness of several cancers. They can also interact with RNA-binding proteins and be involved in protein translation [96]. The serum levels of circ_SETDB1 were associated with ovarian cancer progression, metastases, and primary chemoresistance [97]. In another study, it was found that ovarian cancer platinum sensitivity could be regulated by circ_Cdr1as sponging miR-1270 [98]. In the cisplatin-resistant group of patients, the circulating circ_RNA foxed box protein P1 (circ_FoxP1) was found to be significantly increased and correlated with clinical features of disease advancement and with patients' survival. Circ_FoxP1-mediated sponging of miR-22 and miR-150-3p regulates chemosensitivity, as the use of inhibitors of these two miRNAs enhanced cisplatin resistance. The regulation through sponged miRNAs involves CCAAT enhancer binding protein gamma (CEBPG) and formic-like 3 (FMNL3) proteins engaged in the response to the cellular stress and regulation of cell morphology and cytoskeletal organization, respectively [99].

## 3. PARP Inhibitor Resistance

In PARP inhibitor-sensitive HGSOC cancers, the significantly lower expression of four genes (tyrosine-protein kinase Met or hepatocyte growth factor receptor-*C-MET*, *c*yclin-dependent kinase inhibitor 2A-*CDKN2A*, *N-cadherin*, and P-glycoprotein/ATP binding cassette subfamily B member 1-*P-glyc/ABCB1*) was noted. *C-MET* enhances chemoresistance of human ovarian cancer cells [100], while the *CDKN2A* (*p16*) gene is a candidate for the tumor-suppressor gene [101], N-cadherin protein increases cell metastatic capacity [102], and finally *P-glycoprotein/ABCB1* encodes drug transporter systems and activates multi-drug resistance [103]. Accordingly, the next three genes (sprouty RTK signaling antagonist 2-*SPRY2*, *E-cadherin*, and FA complementation group F-*FANCF*) showed enhanced expression in the PARP inhibitor-sensitive HGSOC tumors. *SPRY2* [104] is significantly down-regulated in ovarian cancer and correlates with poor progression-free (PFS) and overall survival (OS) of patients [105]. E-cadherin is over-expressed in well-differentiated ovarian cancers, while under-expressed E-cadherin is detected in ascites, advanced cancer, and metastases, and is predictive of poor OS [106–108]. *FANCF is* involved in HR-mediated DNA repair and has a possible role in cell response to DNA-damaging agents. *FANCF* suppression plays a role in ovarian cancer occurrence and poor disease outcome [109–111]. The knockdown of *RAD50* in ovarian cancer cell lines improved sensitivity to the PARP inhibitors olaparib and rucaparib [112]. Ovarian cancer cell lines showing *MYC* amplifications were also more sensitive to the PARP inhibitor [112,113]. Patients without BRCA-mutated tumors, whose tumors were BRCA-wild type but had loss-of-function HRR mutations, could have a similar treatment benefit from olaparib-based therapy. These specific HRR mutations identified in HGSOC tumor samples were: DNA repair and recombination proteins-*RAD54L*, *RAD51B*, *RAD54L* rearr (gene rearrangement), *RAD51C*, *RAD52* del (gen deletion), *ATM* rearr, FA complementation group inter-strand DNA cross-link repair protein genes *FANCA* rearr, *FANCD2*, *FANCL* rearr, *FANCL*, *BRIP1*, *CDK12*, and RAD51 paralog *XRCC3* rearr [114].

In another study, the over-expression of the histone methyltransferases EHMT1 and EHMT2 were shown to be responsible for PARP inhibitor resistance in HGSOC [115]. Treatment using PARP inhibitors (PARPi) results in acquired PARPi-resistance, promoted by STAT3 activity both in tumor cells and in immune and CAF cells. Upon PARPi-triggered STAT3 activation, immune cells decrease secretion of interferon-γ and granzyme B and increase secretion of immunosuppressive IL10 cytokine. Treatment of olaparib-resistant ovarian cancer cell line with napabucasin, the STAT3 inhibitor, down-regulated the STAT3 downstream genes, disturbed tumor progression, and improved PARPi sensitivity [116].

## 4. Analysis of Tumor Microenvironment (TME) Biomarkers

### 4.1. Hypoxia and Chemo-Refractory HGSOC

One of the most important stressors originating from tumor TME is hypoxia which regulates cancer aggressiveness, invasiveness, metastatic potential, and chemoresistance through hypoxia-inducible factor-1 alpha (HIF-1α). It is speculated that hypoxia is the key driver of primary chemo-refractoriness of HGSOC [117]. In ovarian cancer samples of non-responders to chemotherapy, the down-regulation of angiogenesis-associated protein angiopoietin-like 4 (ANGPTL4), epidermal growth factor receptor HER3, and HIF-1α was observed [118]. There are also oxygen-independent ways of HIF-1α stimulation, including accumulation of lactate, pyruvate, and succinate [119–121]. Over-expression of HIF-1α is significantly correlated with platinum resistance in ovarian cancer and exposure to hypoxia during the treatment increases the chemoresistance to cisplatin and paclitaxel [118,122]. HIF-1α-mediated increase in vascular endothelial growth factor (VEGF) and epidermal growth factor (EGF) also contributes to ovarian cancer survival, enhanced EMT, and chemoresistance [123,124]. Hypoxia profoundly changes the secretome of HGSOC, especially in the ascitic environment. Interleukin-31 (IL-31) enhances the aggressive and resistant mesenchymal HGSOC phenotype and correlates with adverse outcome [125]. IL-17 stimulates renewal of cancer stem cells, thus enhancing tumorigenesis and chemore-

sistance [126]. Hypoxia alters the function of transcription factors activator protein-1 (AP-1) and nuclear factor kappa light-chain enhancer of activated B cells (NF-κB) in HGSOC through increased IL-8 and tumor necrosis factor-α (TNF-α) secretion, which is followed by reduction of p53 activity and promotes invasive and chemo-resistant phenotype [127,128]. HIF-1α activation is also followed by increased exosome biogenesis, secretion, and transportation [129], as HIF-1α activates Rab22a, an essential protein during exosome secretion [39]. Exosomes are able to transfer signals for the carboplatin resistance from hypoxic to normoxic HGSOC cells [130]. Hypoxia also modulates expression of miRNAs, and miR-181-5p and miR-940 are transported in exosomes secreted from the HGSOC cells to tumor-associated macrophages (TAMs), causing their M2-polarization [131,132]. Several oxidative stress-related genes have been connected to chemoresistance in HGSOC. NF-E2–related factor 2 coded by *Nrf2* gene play a pivotal role in detoxifying and antioxidant defense by transcriptional up-regulation of many downstream genes. However, the *Nrf2* gene can also enhance cancer cells' resistance to anticancer drugs [133]. Other genes from this group include Rac/Cdc42 guanine nucleotide exchange factor 6 (*ARHGEF6*), thioredoxin reductase 1 (*TXNRD1*), alpha-galactosidase A (*GLA*), and glutathione S-transferase zeta 1 (*GSTZ1*) [134].

### 4.2. Therapy-Induced Senescence and Secondary Chemoresistance

Paclitaxel and platinum can cause changes akin to these observed in the response to TME stressors. All of them are indicators of therapy-induced senescence (TIS) and consist in autophagy, metabolic reprogramming, and EMT [117]. TIS describes a molecular and metabolic state of cancer cells that are able to escape dormancy and restore the recurrent tumor with use of acquired adaptations developed in response to the stressors, including previous chemotherapy. Moreover, senescence-associated reprogramming promotes cancer stemness [135,136]. The cancer DNA damage caused by chemotherapy is one of the key inducers responsible for activation of TIS, and defective DNA damage repair (DDR) is a hallmark of advanced and recurrent ovarian cancer [137]. Upon TIS, cancer cells acquire secretome changes called senescence-associated secretory phenotype (SASP) [138]. IL-6 and IL-8 are cytokines that are over-secreted in senescent cancer cells and pro-inflammatory TME and support secondary chemoresistance [40,139]. Aberrant epidermal growth factor receptor (EGFR) signaling stimulates VEGF, survivin, and B cell lymphoma-2 (BCL-2) anti-apoptotic proteins, thus enhancing chemoresistance [140]. Therapy-resistant HGSOC is also characterized with over-secretion of transforming growth factor beta (TGF-β) and increased expression of TGFβ receptor 2 [141,142]. Moreover, dysregulated TGFβ/SMAD family member 4 (SMAD4) signaling pathway may lead to epigenetic silencing of a tumor suppressor RUNX1 partner transcriptional co-repressor 1 (RUNX1T1) [143]. Chemo-resistant HGSOC cells secrete exosomes containing multidrug resistance proteins and increased concentrations of cisplatin [144]. Chemo-resistant HGSOC cells show increased expression of several miRNAs, including miR-93, miR-27a, miR-130a, miR-1246, miR-221, and miR-433, miR-891-5p, miR-200a, and miR-106a, which modulate function of many targets, like phosphatase and tensin homolog deleted on chromosome 10 (PTEN), MDR1, caveolin-1 (CAV-1), cyclin-dependent kinase 6 (CDK6), MYC genes, and B-cell lymphoma/leukemia 10 (BCL-10) [145–153]. Decreased expression of miR-214, miR-30a-5p was noted in platinum-resistant HGSOC, while low expression of miR-216b-5p, and miR-134 was connected to taxane resistance [146,154–156]. Midkine (MK) is a heparin-binding growth factor promoting carcinogenesis and chemo. MK secreted from cancer-associated fibroblasts (CAFs) decreased cisplatin-induced cell death in several cancers, including ovarian cancer cells, and increased the expression of lncRNA ANRIL in the tumor cells. Moreover, ANRIL knockdown in tumor cells restored cisplatin sensitivity [157]. The six CAF-associated genes were linked to chemoresistance and poor outcome of HGSOC patients in another study. These were matrix metalloproteinase *MMP13*, glycoprotein hormones alpha polypeptide *CGA*, ephrin type-A receptor 3 *EPHA3*, proteasome 26S subunit *PSMD9*, paired-like homeodomain 2 *PITX2*, and PH domain leucine-rich repeat protein phosphatase 1 *PHLPP1*, and

were related to CAFs paracrine signaling including MAPK, Ras, and TGF-$\beta$ pathways. The *MMP13*, *CGA*, and *PITX2* enhance cancer cells invasion through stimulation of EMT, TGF-$\beta$ signaling, and gonadotropin secretion. The *EPHA3* influences angiogenesis, metastasis, and inter-cell interactions. The *PSMD9* expression regulates cell resistance to the environmental stressors, and *PHLPP1* acts as a tumor suppressor [158].

TAMs sensitized by the hypoxic TME differentiate into M2-phenotype and secrete exosomes containing miR-223 which are transported into ovarian cancer cells making them chemo-resistant via activation of PTEN/PI3K/AKT signaling pathway. Patients having high levels of circulating exosomal miR-223 had increased chance for ovarian cancer recurrence [159]. Ovarian cancer cells co-cultured with macrophages are able to transfer miR-1246-containing exosomes into M2-type TAMs. The target gene for miR-1246 is a caveolin-1 (*Cav1*) gene involved in exosomal transport. Over-expression of miR-1246 not only was correlated with worse survival prognosis but also with paclitaxel resistance. Oppositely, over-expression of *Cav1* and down-regulation of miR-1246 were able to restore paclitaxel sensitivity [149]. Neutrophils are another component of ovarian cancer TME. Neutrophil extracellular traps (NETs) composed of DNA, histones, myeloperoxidase, elastase, and calprotectin can modulate tumor environment and are involved in the progression and chemoresistance of cancer [160]. The metabolic signatures of ex vivo ovarian cancer cultures comprising amino acids, fatty acids, glutathione, and Krebs cycle pathways enable the discrimination between high and low responders to carboplatin/paclitaxel treatment [161].

### 4.3. Ascites and Chemoresistance

The increased number of spheroids in the ascites of HGSOC patients was correlated with chemoresistance [162]. Similarly, increased numbers of OCT4+EpCAM+CD44+ ovarian cancer stem cells were found in ascites accompanying the chemo-resistant tumors [162]. High levels of insulin-like growth factor (IGF)-I in ascites predicted poor response to neoadjuvant chemotherapy in HGSOC [163]. The high concentrations of cholesterol in ascites stimulate the efficacy of multi-drug resistance protein 1 (MDR1) and ATP binding cassette subfamily G member 2 (ABCG2) efflux pump systems and up-regulate the LXR$\alpha$/$\beta$ cholesterol receptor resulting in enhancement of chemoresistance [164]. Autoantibodies against the tumor-associated antigens BCL6 co-repressor (BCOR), mitochondrial ribosomal protein L46 (MRPL46), and cAMP-responsive element binding protein 3 (CREB3) were decreased in ascites from platinum-resistant patients [165]. The metabolom of ascites is also changed in HGSOC. In the chemotherapy-resistant ovarian cancer tissues, dihydrothymine was significantly reduced, while in the ascites of the drug-resistant group, 1,25-dihydroxyvitamins D3 and hexadecanoic acid were also significantly reduced. Thus, the metabolom of cancer tissues and ascites could affect the drug response [166–169].

Biomarkers of chemoresistance in HGSOC are presented in Table 1.

**Table 1.** Biomarkers of chemoresistance in HGSOC ovarian cancer.

| Genetic and Proteomic Biomarkers | | |
|---|---|---|
| **Biomarker** | **Results of Testing** | **Reference** |
| *BRCA* reverse mutations | Recognized in 18% of patients with chemo-refractory tumors and in 13% of patients with platinum-resistant tumors | [11] |
| RAD51C, RAD51D, PALB2 reverse mutations | Secondary resistance to platinum- and PARP inhibitor-based therapy | [11,12] |
| *MYC*, *RB1*, *PIK3CA*, *BRCA2* N372H SNP | Copy number variations of *MYC*, *RB1*, and *PIK3CA* were noticed in recurrent cancers, while *BRCA2* N372H polymorphism was noticed in recurrent drug-resistant tumors | [14] |
| BRCA2 | The up-regulation of *BRCA2* was observed in chemo-resistant patients, and the down-regulation of *BRCA2* reduced the DNA repair in ovarian cancer cells, sensitizing them to cisplatin | [30] |
| SNPs: rs4910232(11p15.3), rs2549714(16q23), rs6674079(1q22) | Poor response for first-line platinum-based chemotherapy and unfavorable outcome | [22] |

**Table 1.** *Cont.*

| Genetic and Proteomic Biomarkers | | |
|---|---|---|
| **Biomarker** | **Results of Testing** | **Reference** |
| *MROH1, TMEM249, HSF1* | The highest incidence of copy number variations found in these genes was associated with neo-adjuvant chemoresistance | [41] |
| *TP53* | Genome altered fraction of *TP63* mutation and plasma mutation burden had prognostic value according to chemoresistance | [15] |
| *CCNE1* | Amplification or gain of *CCNE1* is related to chemoresistance and decreased OS | [16] |
| *ADAMTS* | Mutations in *ADAMTS* family were associated with a higher chemosensitivity (100% for mutated vs. 64% for wild-type cases), longer platinum-free duration (21.7 months for mutated vs. 10.1 months for wild-type cases), and with significantly better OS and PFS | [23] |
| *MDR1, GSTpi, BCL-2, SMAD4* | Disturbed expression of these genes exclusively drives to acquired chemoresistance in HGSOC | [10,34–38] |
| *AGGF1, MFAP4* | High expression of these genes was observed in HGSOC and correlated with platinum resistance | [39] |
| *AURKA, CCNA2, MYBL2, ORC1, CDK1, TOP2A, TTK* | The expression of *AURKA, CCNA2, MYBL2,* and *ORC1* affected survival of platinum resistant patients, while expression of *CCNA2, CDK1, ORC1, TOP2A,* and *TTK* affected survival of platinum/taxol-resistant patients | [40] |
| *C-MET, CDKN2A, N-cadherin, P-glyc/ABCB1* | The lower expression of these genes was noted in HGSOC sensitive to PARPi | [90] |
| *SPRY2, E-cadherin, FANCF* | Enhanced expression of these genes was observed in PARPi-sensitive HGSOC tumors | [94] |
| Loss-of-function HRR mutations | Patients with HGSOC tumors without BRCA mutations, but with HRR specific mutations of *RAD, CDK, FANCL,* or *BRIP1* genes reacted well to olaparib therapy | [104] |
| Plasma proteins | Complement C4-A, IgJ chain, clusterin, α-1-antitrypsin, and carbonic anhydrase 1 were up-regulated, and transthyretin, haptoglobin, β-2-glycoprotein, Ig γ-2 chain C region, Ig γ-1 chain C region, complement factor I light chain, Igκ chain C region, complement C3, and apolipoprotein E were down-regulated in the chemoresistant group | [42] |
| DSG2 | Desmoglein-2 was over-expressed in serum of HGSOC patients with chemo-refractory cancer and worse survival | [43] |
| sPD-L1/L2 | Soluble receptor ligands related to reduced OS and platinum resistance | [44] |
| COL11A1 | Collagen type XI alpha 1 chain was found to be a biomarker of chemoresistance and correlated with worse survival Down-regulation of COL11A1-mediated ovarian tumor suppression, chemosensitivity, and better survival, thus suggesting its potential application as a therapeutic target COL11A1 regulates TWIST1 to induce chemoresistance. TWIST1 can potentially be targeted in patients with COL11A1-positive ovarian cancer | [45,69,70] |
| Cells | | |
| **Cell** | **Results of Testing** | **Reference** |
| CD44+/CD24− stem cells | Increased number of these cells correlated with chemoresistance | [46] |
| CD44+/CD117+ stem cells | Related to platinum and paclitaxel resistance | [47] |
| ERCC1+ CTCs | Increased numbers of these cells were independent prognostic factor for platinum resistance, OS, and PFS | [48] |
| PI3Kα+ Twist1+ CTCs | Upon platinum-based chemotherapy, CTCs acquired EMT-like phenotype, characterized by a shift towards PI3Kα and Twist-expressing CTCs, which reflect tumor evolution towards therapy-resistant phenotype | [51] |
| Epigenetic Biomarkers | | |
| **Biomarker** | **Results of Testing** | **Reference** |
| H3K27me3 histone | Significantly lower expression for the H3K27me3 was found in "stem-like cells" and in chemo-resistant HGSOC cell lines | [53] |
| H3K79 histone | H3K79 methylation of multi-drug resistance genes augments the chemoresistance of tumor cells | [54] |
| Hydroxymethylcytosine (5hmC) | Regulates gene expression and DNA methylation. Loss of 5hmC levels was associated with resistance to platinum-based therapy and worse patient survival | [55] |
| SLFN11 | CpG island hypermethylation of promoter for SLFN11 was associated with platinum resistance in HGSOC patients | [58] |

**Table 1.** *Cont.*

| | Epigenetic Biomarkers | |
|---|---|---|
| **Biomarker** | **Results of Testing** | **Reference** |
| miR-454-3p, miR-98-5p, miR-183-5p, miR-22-3p | Discriminate between platinum-sensitive and platinum-resistant HGSOC patients | [61] |
| miR-let7g | Low suppressor miR-let7g tumor levels were associated with chemoresistance, and up-regulation of this miRNA promoted cell cycle arrest, inhibited epithelial-to-mesenchymal transition, and restored the chemosensitivity | [63] |
| miR-23a-3p | Suppresses apoptosis of tumor cells and supports platinum chemoresistance by regulating the apoptotic pathway | [64] |
| miR-139-5p | Restoration of miR-139-5p in chemo-resistant ovarian cancer cell lines increased the sensitivity to cisplatin treatment | [65] |
| miR-411 | Fatty acids transporting protein SLC27A2 can bind to the miR-411 promoter region and change its effects on the drug-transporter *ABCG2* target gene. Down-regulated miR-411 expression contributes to ovarian cancer chemoresistance | [66] |
| miR-493-5p | Induces platinum/PARPi resistance exclusively in *BRCA2*-mutated tumors | [68] |
| miR-21 | Expression was up-regulated in cisplatin-resistant ovarian cancer cells, and its expression was regulated by JNK-1/c-Jun/miR-21 pathway | [70] |
| miR-551b | Through the suppression of *Foxo3* and *TRIM31* tumor suppressors, it promotes chemoresistance of ovarian cancer stem cells | [71] |
| miR-137 | The ROS production and DUOXA1 up-regulation cause the activation of c-Myc/miR-137/EZH2 pathway and enhances platinum resistance | [78] |
| miR-9, miR-30a, miR-211 | Up-regulation was correlated with increased chemosensitivity | [72–74] |
| Panel of 7 HGSOC-derived lncRNAs | Up-regulated RP11-126K1.6, ZBED3-AS1, RP11-439E19.10, and RP11-348N5.7, and down-regulated RNF144A-AS1, GAS5, and F11-AS1 predicted chemosensitivity (AUC > 0.8) LINC01363, AC114401.1, and AL360169.2 indicated chemoresistance, and LINC01018, LINC02636, AC090625.2, and AC084781.1 indicated at least partial chemosensitivity | [80,81] |
| PVT1 lncRNA | Expression of PVT1 was significantly higher in ovarian cancer tissues of cisplatin-resistant patients, and promoted cisplatin resistance by the regulating of the apoptotic pathways | [82] |
| NF-κB/HOTAIR lncRNA pathway | NF-κB-HOTAIR pathway contributes to cellular senescence and chemotherapy resistance | [84] |

*BRCA*—breast cancer antigen; *RAD51C*—RAD51 homolog C; *RAD51D*—RAD51 homolog D; *PALB2*—partner and localizer of BRCA2; *MYC*—Myc family regulatory gene and proto-oncogene; *RB1*—retinoblastoma 1 gene; *PIK3CA*—phosphatidylinositol-4,5-bisphosphate 3-kinase catalytic subunit alpha; SNP—single nucleotide polymorphism; *TP53*—tumor protein p53; *CCNE1*—cyclin E1; OS—overall survival; *ADAMTS*—A disintegrin and metalloproteinase with thrombospondin motifs; PFS—progression-free survival; *MDR1*—multi-drug resistance 1; *GSTpi*—glutathione S-transferase Pi 1; *BCL-2*—B-cell lymphoma 2; *SMAD4*—SMAD family member 4; *AGGF1*—angiogenic factor with G-Patch and FHA domains 1; *MFAP4*—microfibril-associated protein 4; *AURKA*—Aurora A kinase; *CCNA2*—cyclin A2; *MYBL2*—MYB proto-oncogene-like 2; *ORC1*—origin recognition complex subunit 1; *CDK1*—cyclin dependent kinase 1; *TOP2A*—DNA topoisomerase II alpha; *TTK*—threonine tyrosine kinase; *C-MET*—tyrosine-protein kinase Met or hepatocyte growth factor receptor; *CDKN2A*—cyclin dependent kinase inhibitor 2A; *P-glyc/ABCB1*—P-glycoprotein/ATP binding cassette subfamily B member 1; *SPRY2*—sprouty RTK signaling antagonist 2; *FANCF*—FA complementation group F; HRR—homologous recombination repair; *FANCL*—FA Complementation Group L; *BRIP1*—BRCA1 interacting helicase 1; *ROH1*—maestro heat-like repeat-containing protein family member 1; *TMEM249*—transmembrane protein 249; *HSF1*—heat shock transcription factor 1; DSG2—desmoglein-2; sPD-L1/L2—soluble programmed death receptor ligands; COL11A1—collagen type XI alpha 1 chain; TWIST1—twist family basic helix–loop–helix transcription factor 1-related protein 1; ERCC1—excision repair cross-complementation group 1 protein; CTCs—circulating tumor cells; PI3Kα—phosphatidylinositol 3-kinase alpha; Twist 1—Twist-related protein 1; SLFN11—DNA/RNA helicase Schlafen-11; SLC27A2—solute carrier family 27 member 2; PARPi—poly ADP ribose polymerase inhibitors; JNK-1—c-Jun N-terminal protein kinase 1; *Foxo3*—forkhead box O3; *TRIM31*—tripartite motif containing 31; ROS—reactive oxygen species; DUOXA1—dual oxidase maturation factor 1; EZH2—enhancer of zeste 2 polycomb repressive complex 2 subunit methylotransferase; HGSOC—high-grade serous ovarian cancer; lncRNA—long non-coding RNA; PVT1—plasmacytoma variant translocation 1; TGF-*β*—transforming growth factor beta; NF-κB—nuclear factor kappa light-chain enhancer of activated B cells; HOTAIR—HOX transcript antisense RNA.

## 5. Prediction of Chemoresistance—What Are the Needs and What Are the Problems

Primary chemo-refractoriness is a serious obstacle in effective adjuvant and neo-adjuvant treatment of HGSOC; however, this is rather a low-frequency phenomenon. Oppositely, acquired secondary chemoresistance to platinum–taxol chemotherapy is very commonly met in clinical practice and constitutes a deleterious turn in HGSOC therapy.

Similarly, observed resistance to PARP inhibitors has strongly unfavorable clinical consequences. Therefore, early identification of potentially resistant tumors is of the utmost importance for successful therapy. As we could see, many different biomarkers of drug resistance have been identified in tumor tissues, CTCs, ctDNA, and blood plasma. What we really need is to look for reliable and economically acceptable techniques of tumor characterization both before and during the course of therapy in order to obtain the clinical useful information of its evolving drug resistance. Analysis of ctDNA and exosome cargo seems to be the most promising direction. Such liquid biopsies were shown to represent the molecular landscape of the tumor more credibly than locally biopsied tumor samples [157]. Another solution could be the use of a drug-screening system based on the combined ctDNA isolation and patient-derived xenograft model for drug testing [158]. There are, however, some problems connected to HGSOC biology that make difficult to obtain simple solutions. Firstly, the treatment itself enriches the population of HGSOC cells which is less responsive to the therapy due to the reaction to the stress produced by toxic drugs and the privileged selection of cancer stem cells. The cells with a high-stress signature are the precursors of chemo-resistant and relapsing HGSOC populations. Identification of these populations could enable the optimization of subsequent pharmacological interventions in order to attenuate or eliminate them from the tumor [31,107,159]. Secondly, HGSOC tumors are characterized by temporal heterogeneity, which makes personalized therapy a very demanding task. During the longitudinal mutational analysis of HGSOC at the time of diagnosis, at the moment of molecular recurrence, and finally at the moment of clinical recurrence, it was shown that except for *TP53* and *BRCA1/2*, no other gene shared the same specific gene mutation across all three time points [157]. This observation points out the need for repetitive sampling of the tumor or ctDNA/CTCs testing to have the most actual picture of the disease. The third problem is of a technical matter and concentrates on the search of the most effective system of liquid biopsy and technique for genome sequencing of a high quality (optimal material harvest), sensitivity, and specificity that would provide reliable and replicable results. Another very important question concentrates on the issue of if some of the biomarkers of chemoresistance could become novel targets for targeted therapy aiming at the restoration of chemosensitivity. Selective knockdown of some genes or non-coding RNAs, using RNA mimics to change gene expression, and monoclonal antibodies against regulatory proteins are all potential therapeutic ways to overcome chemoresistance. Despite these problems and doubts, it seems that adjustment of the therapy to tumor genomic signatures, TME composition, and chemoresistance profile could be one of the most promising future directions in HGSOC therapy. Such personalization of treatment should replace today unified therapy for all HGSOC tumors. The parallel way which should be developed is identification of HGSOC risk factors and building up a screening strategy, especially for the average-risk population of women. Advances in this field could enable the prophylaxis and diagnosis of HGSOC in less advanced clinical stages (I/II), when therapy is more efficient and cost limited. Emphasis on the early recognition of HGSOC tumors could favorably push the moment of diagnosis from the present 75% of advanced cases towards the majority of early-stage (I/II) tumors. This could further enable the transference of more money towards individualized therapy of advanced and recurrent tumors.

**Author Contributions:** Conceptualization, data collection, writing—original draft preparation, J.W. (Jacek Wilczyński); writing—review and editing, J.W. (Justyna Wilczyńska) and E.P.; review and editing, supervision, M.W. All authors have read and agreed to the published version of the manuscript.

**Funding:** This work was supported by the National Science Centre of Poland, grant No. 2019/33/B/NZ7/02872 (http://www.ncn.gov.pl/, accessed on 8 October 2023).

**Conflicts of Interest:** The authors declare no conflict of interest.

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
