# Peer review of "Prediction of Chemoresistance—How Preclinical Data Could Help to Modify Therapeutic Strategy in High-Grade Serous Ovarian Cancer"

_curroncol, doi:10.3390/curroncol31010015_

Round 1

Reviewer 1 Report

Comments and Suggestions for Authors

In the present manuscript “Prognosis of Chemo-Resistance-Indispensable Element of Modern Therapeutic Strategy in High-Grade Serous Ovarian Cancer”, WilczyÅ„ski et al. highlight the importance to understand and clinically harness chemo-resistance in advanced high-grade serous ovarian cancer (HGSOC). Authors thereby provide a comprehensive review of current preclinical knowledge on potential clinically relevant pathways and biomarkers of resistance to platinum-based chemotherapy.

Altogether, I may congratulate the authors to the comprehensive depiction and broad collection of preclinical data. The gives an overview of relevant up-to-date evidence, correctly stresses current clinical challenges and subsequent preclinical questions of interest, and tells its story with an easy-to-follow read thread.

However, I may provide some thoughts to improve the quality of the present manuscript:

First, I’d like to address some semantic ambiguities:

1.     I’d suggest using “chemoresistance” instead of “chemo-resistance”. I don’t see the point in separating the word with a hyphen and it is also usually not used as a compound in literature. This would increase readability: It took me three attempts to understand how “Chemo-Resistance-Indispensable Element” in the title should read. Probably consider involving a native speaker for English editing.

2.    Eg line 545 “chemo-resistance to PARP inhibitors” ~ even though not de-facto incorrect, the use of “chemotherapy resistance” in literature is usually confined to “classic” cytotoxic chemotherapies, and not “new” targeted therapies. To improve readability, I would suggest to replace “chemo-resistance” whenever used in association with non-cytotoxic chemotherapy related meanings, e.g., “resistance to PARP inhibitors”.

3.    Until the very end, I was not sure if authors actually mean “prognosis of chemo-resistance”. I think it should rather read “prediction of chemo-resistance”. Prognosis and Prediction are fundamentally different and are often misinterpreted to be interchangeable. In terms of medical biostatistics, they are not: The endpoint of a prediction is binary (resistance yes/no, recurrence yes/no), the endpoint of prediction prognosis / to prognosticate is survival = PFS/ OS. I may suggest the authors reconsider the semantics as they are fundamental to this review. Ballmann gives a good overview on this topic: 10.1200/JCO.2015.63.3651 Journal of Clinical Oncology 33, no. 33 (November 20, 2015) 3968-3971.

Content-related points:

-       Line 42: “the 5 year survival in advanced stages has not changed in the last decade…” – this claim does not reflect current evidence, PARPi are standard of care and significantly improved the 5-year OS: I invite to compare e.g. Olaparib plus bevacizumab first-line maintenance in ovarian cancer: final overall survival results from the PAOLA-1/ENGOT-ov25 trial Ray-Coquard, I. et al. Annals of Oncology, Volume 34, Issue 8, 681 – 692: 5- year OS rate in HRD+ disease was 66% with olaparib + bevacizumab versus 48% with placebo + bevacizumab. The featured citation is outdated.

-       Line 50-51: “while in the group of BRCA-mutated patients PARP inhibitors are used..” –the biomarker for PARP induction is HRD, not BRCA mutational status.

-       Line 51-52 “PARPi have shown rather moderate efficacy..” – I strongly disagree with this claim. The introduction of PARPi revolutionized the therapy of HGSOC probably as much as the introduction of platinum salts in the 90s. Phase III trials demonstrated almost doubled PFS rates in most sensitive cohorts. Interestingly, authors did not seem to quote relevant studies here, I’d suggest to cite PAOLA-1, PRIMA, VELIA..

-       Line 549: “What we really need, is to get this knowledge right and make some clinically useful conclusions for the future.” – even though I totally agree with the authors, this is a somewhat sloppy sentence for a scientific paper. I’m convinced the authors can make this point using some more scientific English.

-       Line 576-578: same as above “Despite these problems and doubts it seems that there is no other way than to adjust the lines of therapy to tumor genomic signatures, TME composition and chemo-resistance profile.” – that thought has its merits and appears to be one of the most promising approaches to date, but there is no need to phrase this thought as an absolute - as it is not.

I do not have any questions to raise in terms of the high-quality preclinical review of literature the authors provide. However, considering the present review addresses preclinical data and potential future implications on clinical medicine, I’d strongly suggest to include this information into the title. Moreover, I’m unhappy with the fact that the title appears to suggest that chemo-resistance related personalized medicine already arrived in clinical routine. It has definitely not. The idea is promising, but hypothetical at the moment and lacks prospective clinical validation. Therefore “Indispensable Element of Modern Therapeutic Strategy” is a bold claim, not supported by data. I may suggest to revise the title accordingly.

Author Response

# P.T. Reviewer No 1

We would like to thank you for your positive reception of our manuscript. We also very much appreciate your valuable remarks.

1.     I’d suggest using “chemoresistance” instead of “chemo-resistance”. I don’t see the point in separating the word with a hyphen and it is also usually not used as a compound in literature. This would increase readability: It took me three attempts to understand how “Chemo-Resistance-Indispensable Element” in the title should read. Probably consider involving a native speaker for English editing.

Both the words „chemo-resistance” and „chemo-sensitivity” have been devoid of the hyphen.

2.    Eg line 545 “chemo-resistance to PARP inhibitors” ~ even though not de-facto incorrect, the use of “chemotherapy resistance” in literature is usually confined to “classic” cytotoxic chemotherapies, and not “new” targeted therapies. To improve readability, I would suggest to replace “chemo-resistance” whenever used in association with non-cytotoxic chemotherapy related meanings, e.g., “resistance to PARP inhibitors”.

It has been changed, as well as, „chemosensitivity” into „sensitivity”

3.    Until the very end, I was not sure if authors actually mean “prognosis of chemo-resistance”. I think it should rather read “prediction of chemo-resistance”. Prognosis and Prediction are fundamentally different and are often misinterpreted to be interchangeable. In terms of medical biostatistics, they are not: The endpoint of a prediction is binary (resistance yes/no, recurrence yes/no), the endpoint of prediction prognosis / to prognosticate is survival = PFS/ OS. I may suggest the authors reconsider the semantics as they are fundamental to this review. Ballmann gives a good overview on this topic: 10.1200/JCO.2015.63.3651 Journal of Clinical Oncology 33, no. 33 (November 20, 2015) 3968-3971.

Thank you, it is a very constructive remark, and you are right, should be „prediction”. The appropriate places of the text has been changed.

Content-related points:

-       Line 42: “the 5 year survival in advanced stages has not changed in the last decade…” – this claim does not reflect current evidence, PARPi are standard of care and significantly improved the 5-year OS: I invite to compare e.g. Olaparib plus bevacizumab first-line maintenance in ovarian cancer: final overall survival results from the PAOLA-1/ENGOT-ov25 trial Ray-Coquard, I. et al. Annals of Oncology, Volume 34, Issue 8, 681 – 692: 5- year OS rate in HRD+ disease was 66% with olaparib + bevacizumab versus 48% with placebo + bevacizumab. The featured citation is outdated.

That line has been changed in order to consult the results of PAOLA-1/ENGOT study - Therefore, the 5-year survival in advanced patient population (clinical stages III - IV) is unsatisfactory, although recently has been improved by introduction of poly-ADP-ribose polymerase (PARP) inhibitors in the group of homologous recombination deficiency (HRD)-positive tumors

-       Line 50-51: “while in the group of BRCA-mutated patients PARP inhibitors are used..” –the biomarker for PARP induction is HRD, not BRCA mutational status.

It has been changed - while in the group of HRD-deficient patients the poly-ADP ribose polymerase (PARP) inhibitors are used, with PARP-based therapy showing satisfactory efficacy

-       Line 51-52 “PARPi have shown rather moderate efficacy..” – I strongly disagree with this claim. The introduction of PARPi revolutionized the therapy of HGSOC probably as much as the introduction of platinum salts in the 90s. Phase III trials demonstrated almost doubled PFS rates in most sensitive cohorts. Interestingly, authors did not seem to quote relevant studies here, I’d suggest to cite PAOLA-1, PRIMA, VELIA..

The three mentioned trials have beed cited - In phase III PAOLA-1/ENGOT-ov25 trial patients with the BRCA-mutated tumors had a benefit from maintenance therapy using PARP inhibitor olaparib + bevacizumab versus placebo + bevacizumab (24-month progression-free survival PFS 89% vs. 15%). In VELIA study, complete response rates among patients residual disease after cytoreduction were 24% in PARP inhibitor veliparib group compared to18% in controls. The PRIMA/ENGOT-ov26/GOG-3012 trial showed that in newly diagnosed advanced ovarian cancer  therapy with PARP inhibitor niraparib was effective compared to placebo irrespectively of HRD status (HRD-positive 38% vs. 17%, overall 24% vs. 14%).

-       Line 549: “What we really need, is to get this knowledge right and make some clinically useful conclusions for the future.” – even though I totally agree with the authors, this is a somewhat sloppy sentence for a scientific paper. I’m convinced the authors can make this point using some more scientific English.

The sentence has been appropriately modified - What we really need, is to look for reliable and economically acceptable techniques of tumor characterization both before and during the course of therapy, in order to get the clinical useful information of its evolving drug resistance.

-       Line 576-578: same as above “Despite these problems and doubts it seems that there is no other way than to adjust the lines of therapy to tumor genomic signatures, TME composition and chemo-resistance profile.” – that thought has its merits and appears to be one of the most promising approaches to date, but there is no need to phrase this thought as an absolute - as it is not.

The sentence has been appropriately modified - Despite these problems and doubts it seems that adjustment of the therapy to tumor genomic signatures, TME composition and chemoresistance profile could be one of the most promising future directions in HGSOC therapy.

I do not have any questions to raise in terms of the high-quality preclinical review of literature the authors provide. However, considering the present review addresses preclinical data and potential future implications on clinical medicine, I’d strongly suggest to include this information into the title. Moreover, I’m unhappy with the fact that the title appears to suggest that chemo-resistance related personalized medicine already arrived in clinical routine. It has definitely not. The idea is promising, but hypothetical at the moment and lacks prospective clinical validation. Therefore “Indispensable Element of Modern Therapeutic Strategy” is a bold claim, not supported by data. I may suggest to revise the title accordingly.

The title has been modified - Prediction of Chemoresistance - how preclinical data could help to modify Therapeutic Strategy in High-Grade Serous Ovarian Cancer

Reviewer 2 Report

Comments and Suggestions for Authors

The review manuscript “Prognosis of chemo-resistance - indispensable element of modern therapeutic strategy in high-grade serous ovarian cancerby Jacek WilczyÅ„ski and co-authors to summary tumor-derived genomic, mutational, cellular and epigenetic biomarkers of HGSOC chemo-resistance, as well as, tumor microenvironment-derived biomarkers of chemoresistance, and discusses their possible use in the novel complex approach to the ovarian cancer therapy and monitoring. However, some concerns that must be taken into account before the work can be reconsidered for publication.

Comments

1.      Lane 245: Some microRNA are identified in ovarian cancer tissues such as miR-335 and miR-509-3p should be involved in manuscript.

2.      Table 1: COL11A1 expression has been detected in tumor cells as well as in tumor-associated stromal cells. It overexpression promotes ovarian cancer progression and chemoresistance. Recently report indicated that COL11A1 is a novel therapeutic target for cancer treatment. These references should be added.

3.      Twist1 also regulated by COL11A1. The reference should be added.

4.      Moderate editing of English language required.

Comments on the Quality of English Language

Moderate editing of English language required.

Author Response

# P.T. Reviewer No 2

The review manuscript “Prognosis of chemo-resistance - indispensable element of modern therapeutic strategy in high-grade serous ovarian cancer” by Jacek WilczyÅ„ski and co-authors to summary tumor-derived genomic, mutational, cellular and epigenetic biomarkers of HGSOC chemo-resistance, as well as, tumor microenvironment-derived biomarkers of chemoresistance, and discusses their possible use in the novel complex approach to the ovarian cancer therapy and monitoring. However, some concerns that must be taken into account before the work can be reconsidered for publication.

We would like to thank you for your positive reception of our manuscript. We also very much appreciate your valuable remarks.

1.      Lane 245: Some microRNA are identified in ovarian cancer tissues such as miR-335 and miR-509-3p should be involved in manuscript.

The paragraph describing both miR-335 and miR-509 has been added.

MiR-335 expression level was observed to be reduced in malignant tissue samples, especially omental implants, and was associated with shorter OS and tumor recurrence. Up-regulation of miR-335-5p expression restored the cisplatin sensitivity of ovarian cancer cells through suppressing BCL2L2 anti-apoptotic gene, suggesting the potential of miR-335-5p/BCL2L2 signaling as a therapeutic target to overcome the cisplatin resistance .High collagen type XI alpha 1 (COL11A1) levels were found to be associated with tumor progression, chemoresistance, and poor patient survival. MiR-509 and miR-335 were identified as the candidate miRNAs regulating COL11A1 expression. Treatment of ovarian cancer cells with miR-335 mimics decreased COL11A1 expression and suppressed cell proliferation and invasion, simultaneously increasing the cisplatin sensitivity.

2.      Table 1: COL11A1 expression has been detected in tumor cells as well as in tumor-associated stromal cells. It overexpression promotes ovarian cancer progression and chemoresistance. Recently report indicated that COL11A1 is a novel therapeutic target for cancer treatment. These references should be added.

Information in the table has been updated - Down-regulation of COL11A1 mediated ovarian tumor suppression, chemosensitivity, and better survival, thus suggesting its potential application as a therapeutic target

3.      Twist1 also regulated by COL11A1. The reference should be added.

The appropriate text and the reference number were added both in the miRNA paragraph and Table 1. 

4.      Moderate editing of English language required.

The language has been checked

Reviewer 3 Report

Comments and Suggestions for Authors

Dear Authors,

This review is extensive and offers a good overview of recent research. It's not an easy reading as you present many markers. Maybe inserting some paragraphs instead of having just one big chunck of text might help a bit. Please find my other notes in the pdf file.

Best of luck!

Comments on the Quality of English Language

minor issues. i tried highlighting as many as i could

Author Response

# P.T. Reviewer No 3

This review is extensive and offers a good overview of recent research. It's not an easy reading as you present many markers. Maybe inserting some paragraphs instead of having just one big chunck of text might help a bit. Please find my other notes in the pdf file.

We would like to thank you for your positive reception of our manuscript. We also very much appreciate your valuable remarks. We have corrected the text according to your demands.

1/ the abstract has been condensed to 202 words

2/ the data of cancer.org internet page have been given the reference number

3/ the data of niraparib therapy in BRCAwt patients have been cited

4/ PAOLA, VELIA and PRIMA trials were cited

5/ the asterisks in miRNA paragraph has been removed

6/ the meaning of the shortcuts rears and del has been added

7/ the last column in the Tab. 1 has been narrowed

8/ the words in the text have been corrected according to your suggestions